# Adoption of Transparency and Openness Promotion (TOP) Guidelines across Journals

Inga Patarčić [1,*] 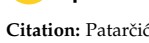 and Jadranka Stojanovski [2,3]

1   Max Delbrück Center for Molecular Medicine, 13125 Berlin, Germany
2   Department of Information Sciences, University of Zadar, 23000 Zadar, Croatia
3   Ruđer Bošković Institute, 10000 Zagreb, Croatia
*   Correspondence: inga.patarcic@mdc-berlin.de

**Abstract:** Journal policies continuously evolve to enable knowledge sharing and support reproducible science. However, that change happens within a certain framework. Eight modular standards with three levels of increasing stringency make Transparency and Openness Promotion (TOP) guidelines which can be used to evaluate to what extent and with which stringency journals promote open science. Guidelines define standards for data citation, transparency of data, material, code and design and analysis, replication, plan and study pre-registration, and two effective interventions: "Registered reports" and "Open science badges", and levels of adoption summed up across standards define journal's TOP Factor. In this paper, we analysed the status of adoption of TOP guidelines across two thousand journals reported in the TOP Factor metrics. We show that the majority of the journals' policies align with at least one of the TOP's standards, most likely "Data citation" (70%) followed by "Data transparency" (19%). Two-thirds of adoptions of TOP standard are of the stringency Level 1 (less stringent), whereas only 9% is of the stringency Level 3. Adoption of TOP standards differs across science disciplines and multidisciplinary journals (N = 1505) and journals from social sciences (N = 1077) show the greatest number of adoptions. Improvement of the measures that journals take to implement open science practices could be done: (1) discipline-specific, (2) journals that have not yet adopted TOP guidelines could do so, (3) the stringency of adoptions could be increased.

**Keywords:** transparency and openness promotion; TOP guidelines; TOP Factor; open science; publishing policies

## 1. Introduction

Science advances knowledge through research and disseminates results via different kinds of research outputs, among which are the most visible scholarly publications. Although scholarly publishing has been witnessing a transition from 'publishing as fast as possible' towards open science practices of 'sharing knowledge as early as possible' [1], many research outputs are stored behind the publisher's paywall, and thus, are inaccessible to a broader audience [2,3]. Recent open science (OS) initiatives call for a fundamental change in how data, materials or results are produced and published, and establish novel practices on how researchers engage and communicate with the public [4]. In general, novel practices in scholarly communication, enabling open access to publications, data, code, methods, educational materials and transparent and open peer review process, have been established to create a more effective and inclusive system of science [5].

Coordinated efforts of publishers, funders, policymakers, institutions, libraries and researchers as a targeted group are required to achieve a greater level of openness in science [6]. Although most scientists agree to embrace disciplinary norms and values of transparency, openness, and reproducibility [7], that was not necessarily the case in practice [8–10]. Furthermore, Baker (2016) showed that "more than 70% of researchers have tried and failed to reproduce another scientist's experiments, and more than half

have failed to reproduce their own experiments" [11]. The benefits of sharing the research outputs are increased transparency and trust, reproducibility and reuse, increased visibility, readability, citation and impact, long-term archiving and preservation, recognition and reputation, and better collaboration opportunities. Still, many scientists are not eager to implement open science norms in their everyday practices if it is left exclusively to their decision and efforts. However, different '(deposit) mandates' have proven to be effective incentives [12]. Thus, along with national/institutional open science policies and funders' requirements [13], the journal's policies and requirements represent one of the key elements of the incentive system that promotes the adoption of open science practices.

In 2015, the Transparency and Openness Promotion (TOP) guidelines were defined in order to become a shared standard for open practices that journals can adopt to promote open science [14]. TOP guidelines consist of eight modular standards that can be used to evaluate journal's policies on data citation, data transparency, material transparency, code transparency, design and analysis, study pre-registration, analysis pre-registration and replication. Standards were defined to have three tiers of increasing stringency-Level 1, Level 2, and Level 3-that move scientific communication toward greater openness; from mentioning specific open science practices towards encouraging, requiring or enforcing them. For example, if journals require researchers to state whether and where code is available, this qualifies them for code transparency standard Level 1. On the other hand, Level 2 code transparency standard demands code to be posted to a trusted repository, whereas Level 3 additionally requires "reported analyses reproduced independently prior to publication" [14,15]. Although similar requirements were made for data and materials transparency standards, other standards have custom-made requirements; such as Level 1 data citation standard is met if a journal clearly describes the citation of data in guidelines.

Since journals can adopt one or more guidelines with different stringencies, in 2020, the Center for Open Science launched the TOP Factor-a quantitative score that reflects a degree of adherence to the TOP guidelines and "Registered reports/Open science badges" interventions [15]. Overall, TOP Factor includes a total of ten subscales (TOP standards plus two effective interventions: "Registered reports" [16] and "Open science badges" [17]), and can go up to 29 which indicates the highest adherence.

Seven years after TOP guidelines were announced, over 5000 journal and funder signatories expressed their support for the guidelines [18] and policies of at least 2000 journals were examined to define a TOP Factor score. Although a few studies reviewed TOP Factors for discipline-specific journals [6,19–22], according to our knowledge, such analysis has not yet been performed in a larger sample of journals and across scientific disciplines. Thus, published studies did not report on how much the adoption of TOP standards differs across scientific disciplines and how well each standard is adopted in general? Likewise, prior to our study, it was unknown which standards are counted in each TOP Factor score. In order to answer those questions, we analyzed TOP Factor scores and individual levels of adoption of TOP guidelines for two thousand journals reviewed by the Center for Open Science.

## 2. Materials and Methods

The Center for Open Science has been evaluating journal policies based on the degree to which they comply with the TOP Guidelines and reporting them as a TOP Factor metric. The latest version of the TOP Factor metric scopes 2000 journals and explains the steps journals are taking to comply with each TOP standard. In other words, the metric provides per journal information on: (1) per standard stringency level (Level 0–3), (2) description of journal's policy that corresponds to each TOP standard, and 3) the TOP Factor. Importantly, nine categories that build a TOP Factor can score from 0 to 3, whereas one category-"Open Science badges"-scores 0 to 2 [23].

We downloaded the TOP Factor (v33, 29 August 2022 3:12 PM) metric [24] and analyzed its content with an in-house R script. First, we extracted levels of stringency required

for each TOP standard across journals and wrote R-scripts that produced figures in the Results section and calculated the mean and median values.

Second, in order to get statistics about the implementation of the TOP guidelines across discipline-specific journals, we extracted information about journal's disciplines from the Scopus content database. We downloaded SCOPUS content coverage [25]. Scopus was selected as a multidisciplinary bibliographic database indexing 27.253 active journals. We selected only the first sheet of the .xlsx file (Scopus Sources May 2022) [25] and imported information about 43,016 active and non-active journals into R. We removed all inactive journals from our analysis and worked with 27.253 active ones. In the Scopus content coverage, journals are categorized into four top-level disciplines of science: life, social, health and physics. However, some journals belong to different combinations of the aforementioned top-level disciplines. We defined such journals as multidisciplinary.

We paired information reported in the Scopus database and TOP Factor metrics by matching the journal's names, or when a name match was not identified, we identified a match based on E-ISSN or P-ISSN identifiers. With such an approach, we managed to match 1824 (91%) journals.

Lastly, to test whether percentages of discipline-specific journals are equal between Scopus content and TOP Factor metric content, we performed Pearson's Chi-squared test using the chisq.test() function in R (number of journals per scientific discipline was used as an input for the test).

Analysis was made in R version 4.1.0. with an in-house R-script deposited in Zenodo [26] and Github [27].

## 3. Results

### 3.1. Most Journals Adopt a Single TOP Standard and Most Standards Are Adopted with Stringency Level 1

We identified a total of 4661 examples of adoption of TOP standards and two additional interventions ("Registered reports" and "Open science badges") in 2000 journals from the TOP Factor metric. In general, identified adoptions were of the stringency Level 1 (less stringent, 67%), followed by Level 2 (N = 1105, 24%) and Level 3 (N = 412, 9%) with median value of 1 and mean = 1.4 (Table 1).

That held true even when we compared individual standards: depending on the standard stringency Level 1 was identified in 62% to 86% of the journals. For seven out of eight TOP standards, Level 3 policies were adopted in less than 6% of journals, however, the "Replication" standard had an adoption across 37% of journals. On the other hand, two interventions-"Registered reports" and "Open science badges"-were generally adopted with stringency Level 3 and Level 2, respectively. Across TOP standards, mean and median levels of adoption were below 1 (with an exception of data citation standard which median of adoption was equal to 1). However, if we filtered out journals that did not implement given standards, we showed that the mean leave of adoption was generally slightly higher than 1, with an exception of replication standard (mean = 1.8, median = 1), "Registered reports" (mean = 2.8, median = 3), and "Open science badges" (mean = 1.9, median = 2)

The majority of journal policies adopted "Data citation" standards (N = 1192, 60%), followed by 45% of journals that adopted "Data transparency", 36% adopted "Design/Analysis reporting guidelines", and 29% adopted "Analysis and Code" transparency (Figure 1). The other four standards were not so frequently adopted: 15% of the journals adopted "Replication", 14% "Materials transparency", 10% "Study pre-registration" and 9% "Analysis plan pre-registration" standards. Likewise, only 10% of the journals required "Registered reports", and 6% issued "Open Science badges".

**Table 1.** Categories of TOP standards and their implementation across journals. Stringency levels were analyzed and reported as the number, mean value and median of journals that adopt a given standard. To obtain statistics for columns 3 and 4, all examples of not implemented TOP standards were filtered out of the analysis. For example, 808 journals that do not implement "Data citation" standards were excluded when mean and median values of the subsample were calculated. To obtain statistics for columns 5 and 6, we calculated the mean and median value across all 2000 journals.

| Category of TOP Standards/Interventions | Number of Journals that Adopt Specific TOP Standard/Intervention (Level 1/Level 2/Level 3) | Mean Level of TOP Standard Adoption (Subsample) | Median Level of TOP Standard Adoption (Subsample) | Mean Level of Adopted TOP Standard (All Journals) | Median Level of Adopted TOP Standard (All Journals) |
|---|---|---|---|---|---|
| Data citation | 1192 (734/446/12) | 1.4 | 1 | 0.8 | 1 |
| Data transparency | 906 (724/158/24) | 1.2 | 1 | 0.6 | 0 |
| Analysis/code transparency | 583 (427/120/36) | 1.3 | 1 | 0.4 | 0 |
| Materials transparency | 274 (181/84/9) | 1.4 | 1 | 0.2 | 0 |
| Design/analysis reporting guidelines | 715 (545/142/28) | 1.3 | 1 | 0.5 | 0 |
| Study pre-registration | 193 (165/22/6) | 1.2 | 1 | 0.1 | 0 |
| Analysis plan pre-registration | 178 (150/22/6) | 1.2 | 1 | 0.1 | 0 |
| Replication | 295 (183/2/110) | 1.8 | 1 | 0.3 | 0 |
| Registered reports | 202 (20/1/181) | 2.8 | 3 | 0.3 | 0 |
| Open Science badges | 123 (15/108/0) | 1.9 | 2 | 0.1 | 0 |
| **Number of policies** | **4661 (3144/1105/412)** | **1.4** | **1** | **0.3** | **0** |

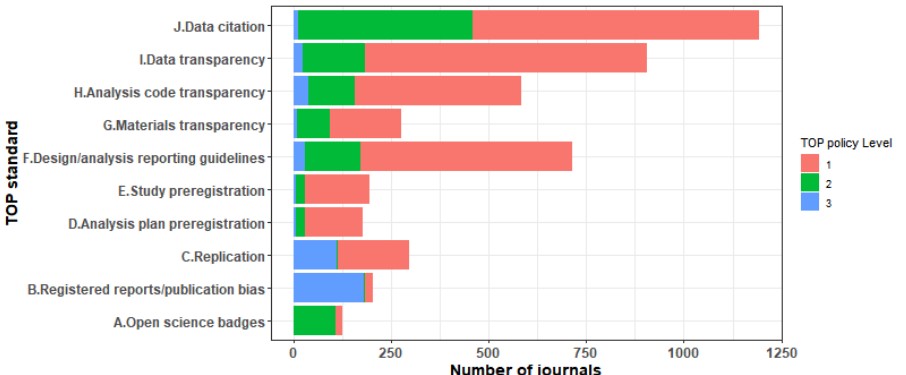

**Figure 1.** Histogram of the number of journals that adopt each individual TOP standard and corresponding stringency level. Stringency levels 1, 2 and 3 of each TOP standard are indicated in different colors, whereas each row is a single TOP standard.

Interestingly, almost one-fourth of the analyzed journal policies did not adopt any of the TOP standards; in other words their TOP Factor score equals to 0 (N = 455, 23%, Figure 2a). A total of 561 journals adopted a single TOP standard (N = 561, Figure 2b, Supplementary Table S1), 70% of which were an adoption of the "Data citation" standard (Supplementary Table S2). In the case of journals that adopt two different standards, the most frequent adoption was of "Data citation" and "Data transparency" standards (47%, N = 123), followed by a combination of "Data citation" and "Design analysis reporting guidelines" (19%, Supplementary Table S3). When three categories of standards were adopted together by a journal, "Data citation", "Data transparency" and "Design analysis reporting guidelines" was adopted in 25% of journals, followed by 21% of journals adopting "Data transparency", "Analysis code transparency" and "Materials transparency" (Supplementary Table S4).

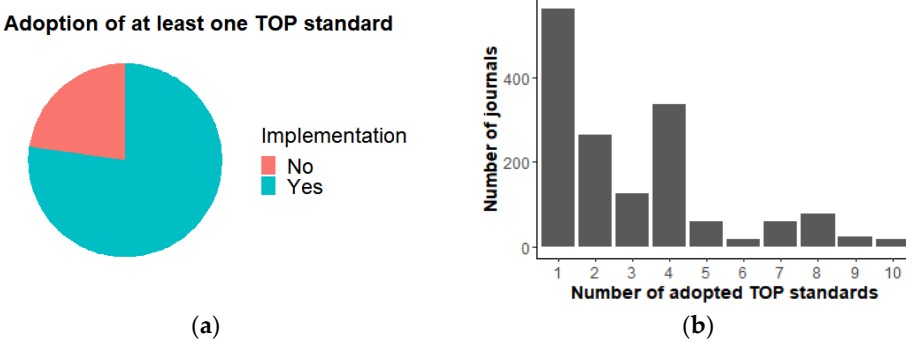

**Figure 2.** Overview of the adoption of TOP standards across 2000 journal policies from the TOP Factor metric. (**a**) Pie chart that shows the number of journals that implement at least one TOP standard: No-indicates journals that do not implement TOP standards (N = 455, TOP Factor = 0), Yes-indicates a number of journals that implement at least one TOP standard (N = 1545, TOP Factor > 0). (**b**) Histogram of the number of journals that adopt one or more TOP standards. An individual journal can adopt up to eight TOP standards, however, two additional interventions-"Open Science badges" and "Registered Reports"-were added to the standards for this plot. Thus, category 10 corresponds to journals that adopt all 8 standards and 2 interventions.

The high proportion of journals adopted four different standards (N = 337); 74% of which correspond to the adoption of "Data citation", "Data transparency", "Analysis code transparency" and "Design analysis reporting guidelines" standards (N = 245, Supplementary Table S5). A total of 78 journals adopt all eight TOP standards, whereas 17 journals issue "Open Science badges" and require "Registered reports" on top of the adoption of all eight TOP standards (Supplementary Table S1).

### 3.2. Adoption of TOP Standards Differ across Disciplines of Science

Journals reviewed in the TOP Factor metric articles are multidisciplinary (35%), and publish articles from social sciences (33%) or other disciplines: "Health", "Life" and "Physical" (16%, 9% and 6% of journals, respectively). However, "active" journals from the Scopus content coverage publish mostly articles from social sciences (32%), followed by 22% of multidisciplinary journals, 21% of "Physical", 17% of "Health" and 7% of journals in the category "Life" (Table 2). These two databases have significantly different content when journal disciplines are considered (Pearson's Chi-squared test X-squared = 328.21 with *p*-value < $2.2 \times 10^{-16}$).

**Table 2.** Number and percentage of discipline-specific journals in the TOP Factor metric and Scopus database.

| Discipline of Science | Number of Journals in the TOP Factor Metric | Share of a Total Number of Journals in the TOP Factor Metric | Number of Journals in Scopus | Share of a Total Number of Journals in the Scopus |
|---|---|---|---|---|
| Social | 595 | 33% | 8799 | 32% |
| Health | 297 | 16% | 4603 | 17% |
| Physical | 118 | 6% | 5848 | 21% |
| Life | 162 | 9% | 1937 | 7% |
| Multidisciplinary | 642 | 35% | 5973 | 22% |
| NA * | 10 | 1% | 93 | 0.34% |

* NA = not available information.

We stratified TOP Factor metrics based on the TOP standards and disciplines of science and showed that multidisciplinary journals have the highest number of adoptions across all eight TOP standards (N = 1418, Supplementary Table S6). "Materials transparency"

standard is an exception since it was adopted by an equal number of multidisciplinary and social sciences journals (N = 81). Journals from the field of social sciences had the second highest number of adoptions of "Data citation" (N = 259), "Data transparency" (N = 220), "Materials transparency" (N = 81), "Study pre-registration" (N = 52), "Analysis plan pre-registration" (N = 42) and "Replication" standards (N = 87). However, a higher number of journals from health sciences (N = 122 & N = 184) than from social sciences (N = 107 & N = 101) adopted "Analysis code transparency" and "Design analysis reporting guidelines", respectively. Only 25 "Physical" journals, as compared to 252 multidisciplinary journals, adopt "Design analysis reporting guidelines".

In terms of shares of a total number of journals, we observed that across all fields of science the "Data citation" standard was most frequently adopted (24–30% of journals), followed by "Data transparency" (18–21%). Interestingly, "Design analysis reporting guidelines" standard was frequently reported in "Health" (22%), multidisciplinary (17%) and "Life" sciences (16%), but was not so frequent in "Social" and "Physical" disciplines (9%). Likewise, four standards: "Materials transparency", "Study pre-registration", "Analysis plan pre-registration" and "Replication" were less frequently, but generally evenly, adopted (3–9%) across disciplines. Issuing of "Open science badges" (N= 31 & N = 55) or requesting "Registered reports" (N = 56 & N = 73) was shown to be done mainly by multidisciplinary and social sciences related journals, whereas, for example, only two "Physical" journals required "Registered reports" (Figure 3).

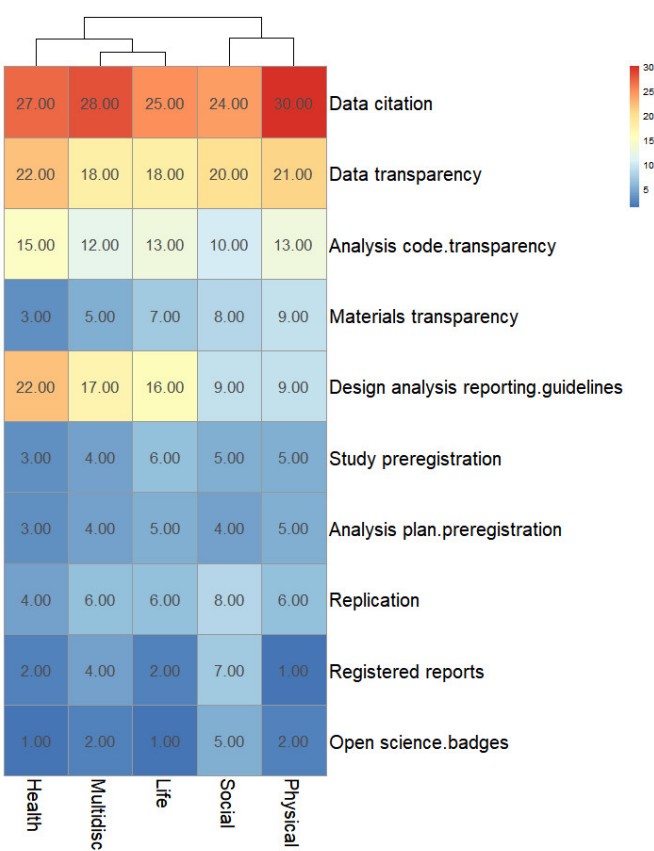

**Figure 3.** Heatmap of the percentage of discipline-specific journals that adopt each individual TOP standard. Statistics are reported per column (e.g., column's values sum up to 100).

## 4. Discussion

An increasing number of journals started adopting a widely appreciated set of TOP guidelines to promote research transparency, openness and reproducibility [6,14,15,19–22]. Journals adopt policies on data citation, data transparency, material transparency, code transparency, design and analysis, study pre-registration, analysis pre-registration and

replication, "in a progressive manner, with policies, such as on the availability of data and code, increasing in strength and rigour over time" [28].

We identified 4661 adoptions of TOP guidelines in 2000 journals, and as expected, the great majority of the journals implement a single TOP standard of the stringency Level 1, where standards are just articulated, stated or described. Although, due to differences in methodology, our results cannot be directly compared to the previous discipline-specific studies [20–22], our finding could provide an explanation for the observed low median values of the TOP Factor. Similarly to the same studies, we identified differences in the number and level of adoption across the TOP standards: "Data citation" and "Data transparency" were the most frequently adopted set of guidelines, thereby rewarding concerned researchers for not receiving more credit for sharing data [29], for the effort they have spent engaging in open practices [14]. In addition, proper data citation supports collaboration and reuse of data, proper attribution and credit and enables reproducibility of findings [30]. However, it remains to be seen what proportion of articles actually report well-formed links to data, data itself and if there is an added value in providing such links [28]. Interestingly, journals frequently adopted the combination of four standards: "Data citation", "Data transparency", "Analysis code transparency" and "Design analysis reporting guidelines", thereby incentivizing openness across all scientific processes: data, code, analysis protocols and design and reducing vague and incomplete reports that decrease confidence in scientific results.

Although standards for pre-registration of studies facilitate the discovery of research, two standards that address pre-registration were more widely adopted by social sciences related journals and less by journals from health and physics disciplines. This is certainly surprising for the field of health, given that application and registration of research are mandatory in many countries (e.g., for trials) and the standard practice of publication of many health journals [31]. Similar was with the category of "Replication" which recognizes the value of replication for independent verification of research results and scientific progress [32]. Our findings indicate that disciplines out of the social sciences can develop their policies in the direction of pre-registration and requesting replication. However, this should be reviewed in the light that journals reported in the TOP Factor metric represent a biased subset of existing science journals, when it comes to the disciplines they cover: for example, journals in the field of physics were significantly underrepresented in our sample. Consequently, these results should be considered cautiously because the distribution of journal disciplines between the "sample of journals" from the TOP Factor metric database and a "population of journals" from the Scopus database differs significantly.

Our study has a set of limitations. Firstly, we did not evaluate policies of the journals ourselves, and thus, we rely on results provided by the COS. Unfortunately, the method of selecting journals included in the TOP Factor by the COS staff or volunteers is not fully transparent, and since we observed a high percentage of journals without any standard in place, or the absence of journals that implement TOP guidelines, we acknowledge but do not fully interpret discipline bias. Therefore, our findings cannot be considered as an objective presentation of journals' transparency and openness policies but only as an analysis of the present coverage of TOP Factor metric.

This study could certainly be expanded by analyzing the level of implementation of specific standards in practice, especially by using the TRUST process [15]. Namely, research has shown that the mere presence of a standard's statement does not mean that it will be respected in practice [33]. Additionally, the appropriateness of the specific TOP Factor standards for different disciplines could be examined. There is already a discussion and developed methodology on this topic published in [15]. Although it was not our study's topic, we also recorded a journal selection bias towards large publishers using common platforms for policies' recommendations and dissemination, which we plan to investigate further in our next study. Additionally, we are planning to add a component of time to get an insight in the evolution of requirements from publishers for adoption of open science practices.

## 5. Conclusions

The majority of the journal policies reported in the TOP Factor metric align to at least one of the TOP's standards, most likely "Data citation" (70%) and with the stringency Level 1. We identified standard-specific and discipline specific differences in implementation of the TOP guidelines that indicated that the improvement of the measures that journals take in order to implement open science practices could be made in three directions: (1) journals that have not yet adopted policies that promote open science could do so, (2) the stringency of the requirements for open science practices for journals who adopted such policies could be increased, and (3) discipline-specific actions could be made. For example, journals from social and physical sciences could more often implement "Design analysis reporting guidelines", whereas "Materials transparency" standards should be more often requested in all disciplines. "Design/analysis reporting guidelines" standards could be more frequently implemented by the journals of physical science, whereas "Registered reports" and "Open Science badges" could be largely deployed by "Life", "Physical", "Health" disciplines, etc. However, since the distribution of journal disciplines between the TOP Factor metric and global distribution of journals according to bibliographic databases differs significantly, these results should be considered with caution.

**Supplementary Materials:** The following supporting information can be downloaded at: https://zenodo.org/record/7361822 (accessed on 25 November 2022), Supplementary Tables S1–S6: Supplementary Table S1. Table of the number of journals that adopt zero to eight TOP standards and two interventions.; Supplementary Table S2. Table of the number and percentage of journals that adopt a single TOP standard.; Supplementary Table S3. Table of the number and percentage of journals that adopt two TOP standards and corresponding combinations.; Supplementary Table S4. Table of the number and percentage of journals that adopt three TOP standards and corresponding combinations.; Supplementary Table S5. Table of the number and percentage of journals that adopt four TOP standards and corresponding combinations.; Supplementary Table S6. Table of the number of journals that implement each TOP standard across different disciplines of science.

**Author Contributions:** Conceptualization, I.P. and J.S.; methodology, I.P. and J.S.; software, I.P.; validation, I.P. and J.S.; formal analysis, I.P.; data curation, I.P.; writing—original draft preparation, I.P.; writing—review and editing, J.S.; visualization, I.P. All authors have read and agreed to the published version of the manuscript.

**Funding:** This research received no external funding.

**Institutional Review Board Statement:** Not applicable.

**Informed Consent Statement:** Not applicable.

**Data Availability Statement:** Publicly available datasets were analyzed in this study. TOP Factor data (v33, 29 August 2022 3:12 PM) can be found at https://osf.io/kgnva/files/osfstorage/5e1350225 7341901c3805317 (accessed on 3 September 2022) whereas Scopus content version (existJuly2022.xlsx) can be downloaded from https://www.elsevier.com/solutions/scopus/how-scopus-works/content? dgcid=RN_AGCM_Sourced_300005030 (accessed on 11 September 2022). The code and data presented in this study are available here: https://zenodo.org/record/7361822 (accessed on 25 November 2022) and can be cited as: [dataset] Inga Patarcic, & Jadranka Stojanovski. 2022. Code & Data for "Adoption of Transparency and Openness Promotion (TOP) guidelines across journals"; Zenodo. Version 4; https://zenodo.org/record/7361822 (accessed on 3 September 2022).

**Conflicts of Interest:** Since J.S. was one of the editors of the Publications Special Issue, the peer review process and editorial decision were performed independently.

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
