# Peer review of "Adoption of Transparency and Openness Promotion (TOP) Guidelines across Journals"

_publications, doi:10.3390/publications10040046_

Round 1

Reviewer 1 Report

The topic is relevant, and empirical studies on transparency and openness of academic journals are welcome. However, this paper has some serious shortfalls that should be fixed before acceptation.

(1) Introduction: The statement that studies on the TOP Factor are missing is simply wrong. A simple search with Google Scholar will retrieve a growing number of empirical papers in different fields. This must be fixed: make a careful literature review and add a state of the art section to the paper. One example with an acceptable methodological approach (I am NOT one of the authors, there is no conflict of interest...) is here Principles of open, transparent and... | Wellcome Open Research And there are others.  

(2) Methodology: It must be clearly stated that this is NOT original work but a reanalysis of published data (= the survey data included in the COS database); this is not bad but it should be explicit. The literature review will also show that the adopted methodology is not complete. It is not enough to count journal numbers; in order to improve the results and to make them comparable with other studies, mean and median statistics should be added for the whole sample, for each TOP criteria (standard) and for each discipline (domain).

(3) Discussion: The authors discuss (quite correctly) three aspects: sample bias, disciplinary bias, and the role of academic publishers. (a) Sample bias: the authors should re-read the documentation on the TOP Factor site to better understand the journal selection and the potential bias; it is (more than) probable that the real adoption of the TOP standards is surestimated because journals without any research data policy will be less represented here than those with such a policy. (b) Disciplinary bias: based on mean and median statistics (and not only on journal numbers), this discussion should be revised. (c) The role of academic publishers: the authors identified the impact of what they call "prestigious commercial publishers". In fact (we did the same kind of research with journals in SSH), the independent variable is not prestige but the dissemination or not via large, often multidisciplinary platforms (Springer, Elsevier, Taylor & Francis etc.) which propose and recommend a standard data policy to their journals.

One aspect is missing in the discussion: the question if all these TOP Factor standards are appropriate in the same way to all disciplines. The challenges and risks are not the same, the way of doing research is not the same, the produced data is not the same. The COS already started this discussion, see https://doi.org/10.1186/s41073-021-00112-8

Author Response

Reviewer 1

Reviewer 1: (1) Introduction: The statement that studies on the TOP Factor are missing is simply wrong. A simple search with Google Scholar will retrieve a growing number of empirical papers in different fields. This must be fixed: make a careful literature review and add a state of the art section to the paper. One example with an acceptable methodological approach (I am NOT one of the authors, there is no conflict of interest...) is here Principles of open, transparent and... | Wellcome Open Research And there are others.

Authors: Thanks for the suggestion. We have added relevant papers and rephrased the introduction.

Reviewer 1: (2) Methodology: It must be clearly stated that this is NOT original work but a reanalysis of published data (= the survey data included in the COS database); this is not bad, but it should be explicit. The literature review will also show that the adopted methodology is not complete. It is not enough to count journal numbers; in order to improve the results and to make them comparable with other studies, mean and median statistics should be added for the whole sample, for each TOP criteria (standard) and for each discipline (domain).

Authors: Thanks for the suggestion. We rephrased the abstract and discussion accordingly and added information about the mean and median values for the whole sample and each TOP standard in the Table 1 of our results. We addressed discipline bias in our Discussion and, due to it, we opted not to provide that mean and median value per discipline. 

Reviewer 1: (3) Discussion: The authors discuss (quite correctly) three aspects: sample bias, disciplinary bias, and the role of academic publishers. (a) Sample bias: the authors should re-read the documentation on the TOP Factor site to better understand the journal selection and the potential bias; it is (more than) probable that the real adoption of the TOP standards is surestimated because journals without any research data policy will be less represented here than those with such a policy. (b) Disciplinary bias: based on mean and median statistics (and not only on journal numbers), this discussion should be revised. (c) The role of academic publishers: the authors identified the impact of what they call "prestigious commercial publishers". In fact (we did the same kind of research with journals in SSH), the independent variable is not prestige but the dissemination or not via large, often multidisciplinary platforms (Springer, Elsevier, Taylor & Francis etc.) which propose and recommend a standard data policy to their journals.

Authors: Thank you for this remark. We changed the manuscript accordingly.

Reviewer 1: One aspect is missing in the discussion: the question if all these TOP Factor standards are appropriate in the same way to all disciplines. The challenges and risks are not the same, the way of doing research is not the same, the produced data is not the same. The COS already started this discussion, see https://doi.org/10.1186/s41073-021-00112-8

Authors: We addressed this in the manuscript. Thank you.

Reviewer 2 Report

Thanks for the chance to review this paper. I think it fills a gap and can be influential (hopefully) in improving journals' support of open practices.

There are two things that could be changed. They are not essential, but I think they could increase the impact of the paper.

1. You could explain the TOP levels of stringency (perhaps with a diagram?). You refer to examples but I think it helps the reader if we know about these levels before the analysis happens.

2. Conclusions: the paper could be a bit more prescriptive - maybe recommendations to journals a bit more strongly formulated - a call to action for journals (or academics who are editors and may be able to have some influence) could be made, with maybe outlining benefits for journals, too. 

Author Response

Reviewer 2

Reviewer 2: 1. You could explain the TOP levels of stringency (perhaps with a diagram?). You refer to examples but I think it helps the reader if we know about these levels before the analysis happens.

Authors: Thank you for your suggestion. In order to improve the clarification of TOP guidelines and TOP Factor we rewrote our introduction and added the sentence in Materials & Methods section.

Reviewer 2: 2. Conclusions: the paper could be a bit more prescriptive - maybe recommendations to journals a bit more strongly formulated - a call to action for journals (or academics who are editors and may be able to have some influence) could be made, with maybe outlining benefits for journals, too. 

Authors: Thank you for your suggestion. However, due to limitations of the study (discipline bias), we decided to refrain from being more prescriptive.

Reviewer 3 Report

The article is certainly worth publishing. Only a few non-committal suggestions were included in the text of the paper. Authors should consider them, without obligation to adopt them.

Author Response

Reviewer 3:The article is certainly worth publishing. Only a few non-committal suggestions were included in the text of the paper. Authors should consider them, without obligation to adopt them.

Authors: Thank you for your comments. We adopted the changes.

Round 2

Reviewer 1 Report

Thank you for the reviewed version which provides an appropriate answer to the comments and suggestions.